# WHAT FACTORS AFFECT LLMS AND RLLMS IN FINANCIAL QUESTION ANSWERING?

## ABSTRACT

Recently, the development of large language models (LLMs) and reasoning large language models (RLLMs) have gained considerable attention from many researchers. RLLMs enhance the reasoning capabilities of LLMs through Long Chain-of-Thought (Long CoT) processes, significantly improving the performance of LLMs in addressing complex problems. However, there are few works that systematically explore what methods can fully unlock the performance of LLMs and RLLMs within the financial domain. To investigate the impact of various methods on LLMs and RLLMs, we utilize five LLMs and three RLLMs to assess the effects of prompting methods, agentic frameworks, and multilingual alignment methods on financial question-answering tasks. Our research findings indicate: (1) Current prompting methods and agent frameworks enhance the performance of LLMs in financial question answering by simulating Long CoT; (2) RLLMs possess inherent Long CoT capabilities, which limits the effectiveness of conventional methods in further enhancing their performance; (3) Current advanced multilingual alignment methods primarily improve the multilingual performance of LLMs by extending the reasoning length, which yields minimal benefits for RLLMs. Additionally, we discuss strategies for enhancing the performance of LLMs and RLLMs in financial question answering, which may serve as a inspiration for future improvements. We hope that this study can serve as an important reference for LLMs and RLLMs in the field of financial question answering.

## 1 INTRODUCTION

Recently, large language models (LLMs) have significantly advanced the field of natural language processing (NLP), and more and more researchers utilize LLMs to solve complex tasks in various domains (Zhou et al., 2023; Nie et al., 2024; Qin et al., 2024; Chang et al., 2024; Wang et al., 2025a). Furthermore, some researchers propose prompting-based and agentic frameworks to enhance the capabilities of LLMs (Wei et al., 2022; Yao et al., 2023a; Zhang et al., 2024b; Yao et al., 2023b; Li et al., 2023; Hong et al., 2024), which further expands the capabilities of LLMs in tasks in different fields. Particularly, to enhance the reasoning capabilities of LLMs in addressing complex problems, researchers introduce reasoning large language models (RLLMs). The application of the Long Chain-of-Thought (Long CoT) fully unlocks the understanding ability of RLLMs (Chen et al., 2025). Long CoT entails deeper reasoning, reflective analysis, and a more extensive exploration of logical structures (Gandhi et al., 2025), which enables RLLMs to perform on par with experts in certain domains.

Researchers have explored the application of LLMs in financial question answering. Wu et al. (2023) present BloombergGPT, a 50 billion parameter LLM for financial domain. Fatemi & Hu (2024) propose a multi-agent framework to enhance the performance of LLMs in financial question answering. Xie et al. (2024) introduce a novel financial benchmark, FinBen, including 42 datasets spanning 24 financial tasks. Srivastava et al. (2024) evaluate LLMs in four financial tabular question-answering datasets. Xue et al. (2024) build a benchmark for financial multilingual multi-modal question answering named FAMMA, which is a challenging benchmark for multi-modal LLMs. Wang et al. (2025c) propose FinSage, which solves the financial filings question answering task through retrieval augmented generation (RAG).

Although LLMs have made significant progress in the field of financial question answering, existing work still mainly focuses on enhancing their performance within this domain. With the advancement of LLMs and RLLMs, the factors influencing their financial question-answering capabilities remain under-explored. Inspired by this gap, we systematically investigate the following issues: ($i$) the impact of prompting methods on LLMs/RLLMs; ($ii$) the influence of agentic frameworks on LLMs/RLLMs; and ($iii$) whether multilingual alignment methods can enhance the multilingual financial question-answering abilities of LLMs/RLLMs. We utilize five LLMs and three RLLMs, conducting detailed experiments across seven representative methods.

The mainly novel insights as follows:

- **Current prompting methods and agent frameworks enhance the performance of LLMs in financial question answering by simulating Long CoT.** Effective prompting methods and agent frameworks primarily simulate Long CoT, enhancing LLM performance through extended reasoning lengths. This parallels the performance gains achieved by RLLMs through Long CoT, demonstrating that Long CoT represents a significant bottleneck for the current performance improvements of LLMs.

- **RLLMs possess inherent Long CoT capabilities, which limits the effectiveness of conventional methods in further enhancing their performance.** Since RLLMs possess Long CoT capabilities, the conventional methods that are effective for LLMs cannot further enhance RLLM performance. We speculate that the key to improving RLLMs in the future lies in the introduction of more complex agent mechanisms. This would enable RLLMs to achieve excellent performance through deep reasoning while also standardizing previous outputs via the agent framework mechanisms.

- **Current advanced multilingual alignment methods primarily improve the multilingual performance of LLMs by extending the reasoning length, which yields minimal benefits for RLLMs.** Current multilingual alignment methods depend on the inherent multilingual capabilities of the model, primarily by extending reasoning lengths to create mechanisms similar to Long CoT. This approach mirrors the gains observed with prompting methods and agentic frameworks in LLMs, thus it is not effective for RLLMs. Consequently, it becomes challenging for RLLMs to achieve further performance enhancements in multilingual contexts.

## 2 BACKGROUND

Financial question answering involves inputting a question along with relevant context, resulting in the model generating the answer to the question. Specifically, for the multiple-choice question, the input comprises the question $\mathcal{Q}$, the available options $\mathcal{O} = \{l_1, l_2, ..., l_n\}$, and the context $\mathcal{C}$, while the output consists of the selected options $\mathcal{A}, \mathcal{A} \in \mathcal{O}$. For the open-ended question, the input includes the question $\mathcal{Q}$ and the context $\mathcal{C}$, with the output being the generated answer $\mathcal{A}$. These two types can be defined as follows:

$$\mathcal{A}_{choice} = \underset{choice}{\mathrm{Model}}\left(\mathcal{Q}, \mathcal{O}, \mathcal{C}\right), \tag{1}$$

$$\mathcal{A}_{open} = \underset{open}{\mathrm{Model}}\left(\mathcal{Q}, \mathcal{C}\right), \tag{2}$$

where $\underset{choice}{\mathrm{Model}}\left(\cdot\right)$ presents the model for the multiple-choice question, $\underset{open}{\mathrm{Model}}\left(\cdot\right)$ presents the model for the open-ended question.

## 3 EXPLORATION

To thoroughly investigate the impact of various methods on the performance of LLMs and RLLMs in financial question answering, this study examines three perspectives: prompting methods (§ 3.1), agentic frameworks (§ 3.2), and multilingual alignment methods (§ 3.3).

### 3.1 EXPLORATION OF PROMPTING METHODS

Current research extensively demonstrates that different prompting methods can significantly alter the performance of LLMs (Wei et al., 2022; Wang et al., 2023; Zhang et al., 2024b). The prompting

method modifies certain aspects of the thinking prompt while ensuring the question remains intact, enabling LLMs to solve the question by following the structured steps of the thinking prompt.

We select three representative prompting methods to investigate the impact of varying prompt word changes on the performance of LLMs and RLLMs: Direct, Zero-shot CoT (Kojima et al., 2022), and Plan-and-Solve (Wang et al., 2023). These prompting methods can be defined as:

- **Direct**: The direct prompting method enables LLMs to answer questions by inputting them directly, without the inclusion of the additional thinking prompt. This approach allows for an intuitive assessment of LLM performance and minimizes interference from redundant prompts.

- **Zero-shot CoT**: The zero-shot CoT prompting employs the prompt "let's think step by step" to stimulate the reasoning abilities of LLMs, enabling them to produce longer reasoning processes and, consequently, enhancing their performance.

- **Plan-and-Solve**: The Plan-and-Solve prompting introduces a novel prompt "Let's first understand the problem and devise a plan to solve the problem. Then, let's carry out the plan to solve the problem step by step." This addresses computational errors and enhances the quality of generated reasoning steps through a two-part decomposition.

## 3.2 EXPLORATION OF AGENTIC FRAMEWORKS

To further enhance the performance of LLMs, researchers have started to incorporate agentic systems. Specifically, by enabling LLMs to assume different roles and engage in continuous communication and cooperation, their performance can be significantly improved (Luo et al., 2025a).

We select two agentic frameworks, and aim to explore the impact of LLMs and RLLMs in different agent collaborations: Self-Refine (Madaan et al., 2023) and $S^3$ Agent (Wang et al., 2024). These agentic frameworks can be defined as:

- **Self-Refine**: Self-Refine provides feedback on the output using the same LLMs, thereby iteratively modifying the response and ultimately optimizing the answer to enhance performance. To save cost, we only use one iteration.

- **$S^3$ Agent**: $S^3$ Agent proposes three agents of different perspectives: superficial expression, semantic information, and sentiment expression, to unlock the power of LLMs. We made minor modifications to the $S^3$ Agent for financial question-answering tasks while preserving the three core components intact.

## 3.3 EXPLORATION OF MULTILINGUAL ALIGNMENT METHODS

Due to the varying proportions of different languages in the training corpus, the performance of LLMs typically differs across languages. Improving LLM performance in underrepresented languages has consistently been a key area of research. Current multilingual alignment methods primarily enhance the multilingual capabilities of LLMs through explicit prompts or translation.

We evaluate the performance of four standard multilingual approaches, including state-of-the-art cross-lingual prompting methods, on multilingual alignment: Direct, Translate-en (Shi et al., 2023), and Cross-lingual Prompting (Qin et al., 2023).

- **Direct**: Direct prompting utilizes the English prompt and the local language question, which directly assess the multilingual capabilities.

- **Translate-en**: Translate-en employs translation to convert the local language question into English, thereby preventing performance degradation of LLMs due to insufficient multilingual capabilities.

- **Cross-lingual Prompting (CLP)**: CLP enhances the reasoning abilities of LLMs on multilingual questions through a two-stage multilingual alignment based on prompts: (1) cross-lingual alignment prompting and (2) task-specific solver prompting.

| Method | Arithmetic | | | | Non-Arithmetic | | | | Overall |
|---|---|---|---|---|---|---|---|---|---|
| | Overall | Easy | Medium | Hard | Overall | Easy | Medium | Hard | |
| Llama-3.1-8B-Instruct (Grattafiori et al., 2024) | | | | | | | | | |
| Direct (Grattafiori et al., 2024) | 10.88 | 12.66 | 9.89 | 9.28 | 23.44 | 22.45 | 32.10 | 20.27 | 16.50 |
| Zero-shot CoT (Kojima et al., 2022) | 13.20 | 16.15 | 10.95 | 11.07 | 31.72 | 32.62 | 34.73 | **29.95** | 21.49 |
| Plan-and-Solve (Wang et al., 2023) | 16.09 | 18.99 | 14.84 | 13.17 | 29.77 | 32.62 | 30.52 | 27.92 | 22.21 |
| Self-Refine (Madaan et al., 2023) | 17.95 | 22.70 | 13.07 | 15.56 | **32.06** | 34.74 | 35.26 | 29.27 | 24.26 |
| S$^3$ Agent (Wang et al., 2024) | **19.62** | **22.27** | **15.54** | **19.46** | 30.80 | **37.28** | 32.63 | 26.57 | **24.62** |
| GPT-4o-mini (Hurst et al., 2024) | | | | | | | | | |
| Direct (Hurst et al., 2024) | 32.46 | 41.70 | 30.38 | 21.55 | 42.52 | 50.42 | 50.52 | 34.90 | 36.96 |
| Zero-shot CoT (Kojima et al., 2022) | 34.13 | 41.04 | **34.27** | 24.55 | 45.17 | 56.35 | **52.10** | 36.26 | 39.07 |
| Plan-and-Solve (Wang et al., 2023) | **36.18** | **44.97** | 33.92 | **26.04** | **45.74** | **56.77** | 51.05 | **37.61** | **40.46** |
| Self-Refine (Madaan et al., 2023) | 33.30 | 41.26 | 33.21 | 22.45 | 42.06 | 52.11 | 45.78 | 35.13 | 37.22 |
| S$^3$ Agent (Wang et al., 2024) | 30.88 | 38.42 | 28.26 | 22.75 | 41.72 | 51.69 | 46.84 | 34.23 | 35.73 |
| Gemini-1.5-flash (Team et al., 2024) | | | | | | | | | |
| Direct (Team et al., 2024) | 37.20 | **47.16** | 35.33 | 25.14 | 45.28 | 58.05 | 49.47 | 36.71 | 40.82 |
| Zero-shot CoT (Kojima et al., 2022) | 37.39 | 46.72 | 36.04 | 25.74 | 46.09 | **58.89** | 51.57 | 36.93 | 41.28 |
| Plan-and-Solve (Wang et al., 2023) | **37.76** | 46.50 | **39.57** | 24.25 | **47.12** | 55.50 | **55.26** | **39.18** | **41.95** |
| Self-Refine (Madaan et al., 2023) | 34.41 | 42.35 | 33.92 | 23.95 | 42.52 | 52.96 | 50.00 | 33.78 | 38.04 |
| S$^3$ Agent (Wang et al., 2024) | 37.20 | 45.85 | 34.62 | **27.54** | 44.59 | 57.20 | 50.52 | 35.36 | 40.51 |
| Qwen-2.5-32B (Yang et al., 2024) | | | | | | | | | |
| Direct (Yang et al., 2024) | 41.86 | 50.65 | 40.98 | 30.53 | 48.62 | 59.32 | 56.31 | 39.63 | 44.88 |
| Zero-shot CoT (Kojima et al., 2022) | 42.32 | 50.21 | 40.98 | **32.63** | **50.80** | 59.32 | **60.00** | 42.34 | **46.11** |
| Plan-and-Solve (Wang et al., 2023) | 39.44 | 44.97 | **43.46** | 28.44 | 49.77 | **60.59** | 55.78 | 41.44 | 44.06 |
| Self-Refine (Madaan et al., 2023) | 41.58 | 49.56 | 42.75 | 29.64 | 49.65 | 57.62 | 54.73 | **43.24** | 45.19 |
| S$^3$ Agent (Wang et al., 2024) | **42.51** | **51.52** | 41.34 | 31.13 | 48.85 | **61.44** | 56.31 | 38.96 | 45.34 |
| DeepSeek-V3 (Liu et al., 2024) | | | | | | | | | |
| Direct (Liu et al., 2024) | **56.27** | **65.50** | 58.65 | 41.61 | 62.06 | **73.72** | 69.47 | 52.70 | **58.86** |
| Zero-shot CoT (Kojima et al., 2022) | 56.18 | 63.97 | **59.71** | 42.51 | 61.83 | 71.18 | 70.52 | 53.15 | 58.71 |
| Plan-and-Solve (Wang et al., 2023) | 55.16 | 63.97 | 56.18 | 42.21 | **63.33** | 70.76 | **71.05** | **56.08** | 58.81 |
| Self-Refine (Madaan et al., 2023) | 53.86 | 61.79 | 55.83 | 41.31 | 59.54 | 69.06 | 64.73 | 52.25 | 56.40 |
| S$^3$ Agent (Wang et al., 2024) | 54.69 | 61.57 | 56.89 | **43.41** | 59.42 | 71.18 | 66.31 | 50.22 | 56.81 |
| DeepSeek-R1-Distill-Qwen-32B (Guo et al., 2025) | | | | | | | | | |
| Direct (Guo et al., 2025) | 50.41 | 60.48 | 50.17 | 36.82 | 57.12 | 65.25 | 64.73 | **49.54** | 53.41 |
| Zero-shot CoT (Kojima et al., 2022) | **51.25** | **60.91** | **51.94** | 37.42 | 56.55 | 68.64 | 63.15 | 47.29 | 53.62 |
| Plan-and-Solve (Wang et al., 2023) | 51.16 | **60.91** | 50.53 | 38.32 | 55.86 | 66.52 | 62.63 | 47.29 | 53.26 |
| Self-Refine (Madaan et al., 2023) | 44.65 | 52.18 | 47.70 | 31.73 | 52.06 | 57.20 | 60.00 | 45.94 | 47.96 |
| S$^3$ Agent (Wang et al., 2024) | **51.25** | 59.60 | **51.94** | **39.22** | **58.04** | **69.49** | **65.26** | 48.87 | **54.29** |
| Qwen-3-14B (Yang et al., 2025) | | | | | | | | | |
| Direct (Yang et al., 2025) | 54.88 | 66.59 | **58.65** | 35.62 | 59.08 | **72.03** | **70.52** | 47.29 | 56.76 |
| Zero-shot CoT (Kojima et al., 2022) | **55.72** | **68.34** | 56.18 | **38.02** | **59.19** | 69.91 | **70.52** | **48.64** | **57.27** |
| Plan-and-Solve (Wang et al., 2023) | 54.97 | 66.81 | **58.65** | 35.62 | 57.81 | 71.18 | 68.94 | 45.94 | 56.24 |
| Self-Refine (Madaan et al., 2023) | 54.41 | 65.93 | 56.89 | 36.52 | 58.27 | 71.61 | 66.84 | 47.52 | 56.14 |
| S$^3$ Agent (Wang et al., 2024) | 54.13 | 65.06 | 57.24 | 36.52 | 58.50 | 71.18 | 67.36 | 47.97 | 56.09 |
| O4-mini (OpenAI, 2025) | | | | | | | | | |
| Direct (OpenAI, 2025) | 61.30 | **70.08** | **65.01** | 46.10 | 70.22 | 77.54 | 75.78 | 63.96 | 65.29 |
| Zero-shot CoT (Kojima et al., 2022) | **62.60** | 69.86 | 64.66 | **50.89** | 71.37 | 77.11 | **77.89** | 65.54 | **66.52** |
| Plan-and-Solve (Wang et al., 2023) | 61.67 | 69.86 | 64.31 | 48.20 | **72.41** | **77.96** | **77.89** | **67.11** | 66.47 |
| Self-Refine (Madaan et al., 2023) | 61.20 | 69.21 | 60.77 | 50.59 | 70.34 | 75.00 | 74.73 | 65.99 | 65.29 |
| S$^3$ Agent (Wang et al., 2024) | 61.76 | 69.65 | 62.19 | 50.59 | 71.49 | 76.69 | 75.78 | 66.89 | 66.11 |

Table 1: The results of prompting and agentic methods. **Bold number** presents the best result for these methods on the current model. Underline number presents the second-best result for these methods on the current model. Light yellow presents the current model is LLM, Light green presents the current model is RLLM.

# 4 EXPERIMENTS

## 4.1 EXPERIMENTAL SETTINGS

We use the standard financial question answering benchmark FAMMA (Xue et al., 2024) for experiments. FAMMA is a benchmark for financial multilingual multi-modal question answering, which includes English, Chinese, and French. Since current RLLMs are unimodal and cannot process multimodal information, we utilize the Basic Txt dataset in FAMMA, which converts multimodal data into textual information using OCR. The final dataset used for evaluation comprises 1,945 entries.

We conduct experiments on seven backbones, including five LLMs and three RLLMs: Meta-Llama-3.1-8B-Instruct (Grattafiori et al., 2024), GPT-4o-mini (Hurst et al., 2024), Gemini-1.5-flash (Team et al., 2024), Qwen-2.5-32B (Yang et al., 2024), DeepSeek-V3 (Liu et al., 2024), DeepSeek-R1-Distill-Qwen-32B (Guo et al., 2025), thing mode of Qwen-3-14B (Yang et al., 2025), O4-mini (OpenAI, 2025). Detailed settings can be found in the Appendix A.1.

| Performance | Direct | Translate-en (Shi et al., 2023) | Self-Refine (Madaan et al., 2023) | $S^3$ Agent (Wang et al., 2024) | CLP (Qin et al., 2023) |
|---|---|---|---|---|---|
| Meta-Llama-3.1-8B-Instruct (Grattafiori et al., 2024) | | | | | |
| Chinese | 16.54 | 16.54 (+0.00) | 18.97 (+2.43) | 19.22 (+2.68) | **23.35 (+6.81)** |
| French | 15.38 | 16.66 (+1.28) | 13.46 (-1.92) | **21.15 (+5.77)** | 17.30 (+1.92) |
| Overall (Chinese+French) | 16.22 | 16.57 (+0.35) | 17.46 (+1.24) | 19.75 (+3.53) | **21.69 (+5.47)** |
| GPT-4o-mini (Hurst et al., 2024) | | | | | |
| Chinese | 33.09 | 34.54 (+1.45) | 34.79 (+1.70) | 31.63 (-1.46) | **38.19 (+5.10)** |
| French | 28.84 | 30.76 (+1.92) | 28.84 (+0.00) | 32.69 (+3.85) | **32.05 (+3.21)** |
| Overall (Chinese+French) | 31.92 | 33.50 (+1.58) | 33.15 (+1.23) | 31.92 (+0.00) | **36.50 (+4.58)** |
| Gemini-1.5-flash (Team et al., 2024) | | | | | |
| Chinese | **40.38** | 38.68 (-1.70) | 36.73 (-3.65) | 39.65 (-0.73) | 37.22 (-3.16) |
| French | 27.56 | **29.48 (+1.92)** | 26.92 (-0.64) | 27.56 (+0.00) | **29.48 (+1.92)** |
| Overall (Chinese+French) | **36.86** | 36.15 (-0.71) | 34.03 (-2.83) | 36.33 (-0.53) | 35.09 (-1.77) |
| Qwen-2.5-32B (Yang et al., 2024) | | | | | |
| Chinese | 41.84 | **44.52 (+2.68)** | 44.28 (+2.44) | 42.82 (+0.98) | 43.30 (+1.46) |
| French | 29.48 | 29.48 (+0.00) | 33.97 (+4.49) | **35.89 (+6.41)** | 35.25 (+5.77) |
| Overall (Chinese+French) | 38.44 | 40.38 (+1.94) | **41.44 (+3.00)** | 40.91 (+2.47) | 41.09 (+2.65) |
| DeepSeek-V3 (Liu et al., 2024) | | | | | |
| Chinese | 57.42 | 58.63 (+1.21) | 55.96 (-1.46) | 57.66 (+0.24) | **61.31 (+3.89)** |
| French | 50.00 | 50.00 (+0.00) | 47.43 (-2.57) | 44.23 (-5.77) | **57.69 (+7.69)** |
| Overall (Chinese+French) | 55.37 | 56.26 (+0.89) | 53.61 (-1.76) | 53.96 (-1.41) | **60.31 (+4.94)** |
| DeepSeek-R1-Distill-Qwen-32B (Guo et al., 2025) | | | | | |
| Chinese | 48.41 | 49.87 (+1.46) | 46.47 (-1.94) | **52.06 (+3.25)** | 50.85 (+2.44) |
| French | 49.35 | 42.94 (-6.41) | 44.87 (-1.16) | 49.35 (+0.00) | **52.56 (+3.21)** |
| Overall (Chinese+French) | 48.67 | 47.97 (-0.70) | 46.03 (-2.64) | **51.32 (+2.65)** | **51.32 (+2.65)** |
| Qwen-3-14B (Yang et al., 2025) | | | | | |
| Chinese | **57.66** | 56.69 (-0.97) | 55.23 (-2.43) | 55.47 (-2.19) | 53.77 (-3.89) |
| French | 48.71 | 50.00 (+1.29) | 42.3 (-6.41) | 48.71 (+0.00) | **51.28 (+2.57)** |
| Overall (Chinese+French) | **55.20** | 54.85 (-0.35) | 51.67 (-3.53) | 53.61 (-1.59) | 53.08 (-2.12) |
| O4-mini (OpenAI, 2025) | | | | | |
| Chinese | 65.45 | 65.93 (+0.48) | 64.23 (-1.22) | **67.63 (+2.18)** | 64.54 (-0.91) |
| French | 62.17 | 61.53 (-0.64) | 60.25 (-1.92) | 61.53 (-0.64) | **63.46 (+1.29)** |
| Overall (Chinese+French) | 64.55 | 64.72 (+0.17) | 63.13 (-1.42) | **65.69 (+1.14)** | 64.24 (-0.31) |

Table 2: The results of multilingual alignment methods. **Bold number** presents the best result for these methods on the current model. Light yellow presents the current model is LLM, Light green presents the current model is RLLM. The performance of gains/drops relative to the Direct are highlight with green/red in the Table.

## 4.2 ANALYSIS OF PROMPTING METHODS

**An effective prompting method significantly enhances the performance of LLMs.** As shown in Table 1, advanced prompting method Plan-and-Solve outperforms other prompting and agentic methods on most LLMs. This suggests that employing a well-designed general prompting method effectively enhances the performance of LLMs, even if these methods are not originally proposed for financial question answering.

**Larger LLMs exhibit increased robustness to various prompting methods.** It is evident that for larger LLMs, such as Gemini-1.5-flash, Qwen-2.5-32B, and DeepSeek-V3, the performance gap between different prompting methods is not significant, with performance fluctuations remaining under 2%. This indicates that LLMs with superior performance are less affected by input prompts, demonstrating greater robustness to variations in input.

**For RLLMs, competitive results can be achieved without the use of prompting methods.** Benefiting from the Long CoT process, RLLMs can generate complex reasoning independently, without deliberate guidance from prompting methods. This renders prompting, which is effective for LLMs, less effective for RLLMs and may even diminish their performance. As shown in Table 1, Plan-and-Solve demonstrates poor performance on the three RLLMs.

## 4.3 ANALYSIS OF AGENTIC FRAMEWORKS

**Smaller LLMs can derive greater benefits from the agentic framework.** Smaller LLMs, such as Llama-3.1-8B-Instruct, can significantly improve their performance through the complex agentic framework, which is shown in Table 1. We speculate that this improvement may stem from the relatively weaker ability of smaller LLMs to understand and follow instructions. The standardized agentic framework helps mitigate the probability of the hallucination, thereby further enhancing overall performance.

**A well-designed agentic framework is essential for further enhancing the performance of LLMs.** For larger LLMs, directly employing the agentic framework does not lead to significant performance improvements. This may be due to the fact that these agentic frameworks are not specifically designed for financial question answering, and merely transferring them does not fully enhance LLM performance. To enhance performance in financial question answering, a well-designed agentic framework tailored to this task is necessary.

**The performance gains of RLLMs within agentic frameworks primarily stem from effective agent collaboration.** In RLLMs, agentic frameworks exhibit an improvement compared with prompting methods, particularly in S$^3$ Agent for DeepSeek-R1-Distill-Qwen-32B. This indicates that frameworks successful for LLMs can also be adapted for RLLMs. However, since RLLMs are developed through Long CoT, the performance enhancements associated with agentic frameworks, when employing longer thinking processes, are not pronounced. The benefits of these frameworks are more evident in their inherent design features, such as the reflective capabilities provided by Self-Refine and the multi-faceted thinking fostered by S$^3$ Agent.

## 4.4 ANALYSIS OF MULTILINGUAL ALIGNMENT METHODS

**The Translate-en method can enhance multilingual performance for LLMs to some extent; however, the improvement is not substantial.** As indicated in Table 2, the Translate-en method yields gains across most models compared to the Direct approach, but these gains remain insignificant. This may be attributed to the fact that Translate-en aligns multilingual tasks to a single language. Consequently, the overall reasoning length for LLMs does not increase, resulting in a lack of depth in thinking, which in turn limits the significant enhancement of performance.

**Extending the reasoning length significantly enhances the performance of LLMs.** For most LLMs, employing the CLP method yields better performance than the Translate-en approach. Additionally, the agentic framework (Self-Refine and S$^3$ Agent) enhances the performance of LLMs to some degree. This indicates that CLP not only aligns multiple languages to the target language but also further improves performance by strengthening the Long CoT of LLMs. Since the initial stage alignment performance of CLP is also influenced by the inherent multilingual capabilities of the LLMs, insufficient multilingual ability may limit the performance gains achievable through CLP.

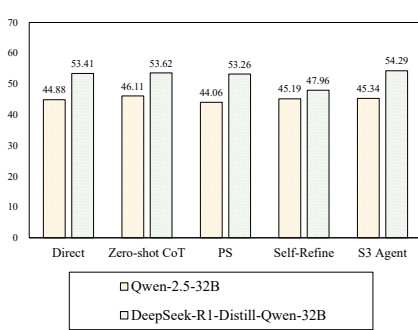

Figure 1: The performance of Qwen-2.5-32B and DeepSeek-R1-Distill-Qwen-32B.

**RLLMs demonstrate self-alignment capabilities for multilingual questions.** In RLLMs, the effectiveness of various alignment methods, such as Translate-en and CLP, is limited. It is posited that RLLMs can achieve results comparable to those of CLP through Long CoT. This suggests that the introduction of CLP may induce overthinking in RLLMs, resulting in a decline in performance. While translation does yield some performance decline, the reduction is not significant.

## 4.5 THE BENEFITS OF LONG COT CANNOT BE OFFSET BY PROMPTING METHODS AND AGENTIC FRAMEWORKS

We compare the performance of Qwen-2.5-32B and DeepSeek-R1-Distill-Qwen-32B, where DeepSeek-R1-Distill-Qwen-32B is a Qwen-2.5 model distilled from DeepSeek-R1, as illustrated in Figure 1. It is evident that DeepSeek-R1-Distill-Qwen-32B outperforms the base model across various methods, demonstrating an average improvement of 7.4%. This indicates that the gains from Long CoT significantly enhance the reasoning capabilities of the base model after RLLMs distillation. Consequently, it is challenging for the base model to surpass RLLMs equipped with Long CoT capabilities through advanced frameworks. While it is possible to narrow this gap using a more complex framework, we argue that this approach may lead to token consumption of LLMs approaching that of RLLMs with the same base model.

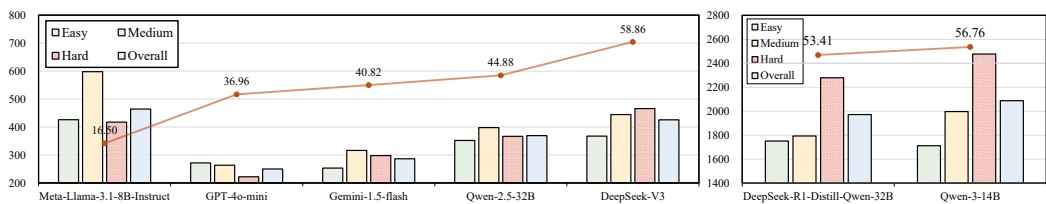

Figure 2: Statistics of average output token consumption for questions of different difficulty in LLMs/RLLMs. The line chart shows the performance of LLMs/RLLMs in the Direct method.

## 4.6 LONGER REASONING PROCESSES GENERALLY LEAD TO BETTER PERFORMANCE

To explore the relationship between output length and performance in LLMs and RLLMs, we conduct a statistical analysis of output tokens. The statistics presented in Table 3 indicate that within the same model, more robust methods require a greater number of tokens. This suggests a positive correlation between performance and token usage in the current LLMs, effective prompting techniques enhance performance by simulating a Long CoT. Statistic of Direct in Figure 2 reveals that, in most LLMs, longer reasoning processes with improved performance. This suggests that sufficient cognitive processing benefits LLMs and their current limitations primarily stem from a constrained thinking process.

| Model | Direct | Zero-shot CoT | Plan-and-Solve |
|---|---|---|---|
| Meta-Llama-3.1-8B-Instruct | 464.36 | 973.27 | 1131.86 |
| gpt-4o-mini | 249.77 | 430.41 | 531.83 |
| gemini-1.5-flash | 286.32 | 296.03 | 313.42 |
| Qwen-2.5-32B | 368.91 | 470.99 | 476.74 |
| DeepSeek-V3 | 425.65 | 450.58 | 493.32 |
| DeepSeek-R1-Distill-Qwen-32B | 1,972.60 | 2006.39 | 2196.63 |
| Qwen-3-14B | 2,087.49 | 2120.97 | 2211.74 |

Table 3: Statistics of average output token consumption in current LLMs/RLLMs.

In contrast, for RLLMs, the output of Long CoT indicates that the length of the CoT is not the primary determinant of performance. For instance, DeepSeek-R1-Distill-Qwen-32B generates an excessive number of tokens for easy questions, yet this does not lead to significant performance enhancements compared with Qwen-3-14B. In easy questions, DeepSeek-R1-Distill-Qwen-32B achieves 53.41%, Qwen-3-14B achieves 56.76% in Acc. This phenomenon may arise from the overthinking in RLLMs (Chen et al., 2024), which fails to yield performance gains while substantially increasing computational costs. Moving forward, dynamically adjusting CoT length according to the complexity of the input problem will represent a critical area of research.

## 4.7 PERFORMANCE IMPROVES AS THE PARAMETERS OF LLMS INCREASE

We further analyze the performance and token consumption of Qwen-3 across various scales and the thinking mode, with the final results illustrated in Figure 3. The data reveal that the performance of

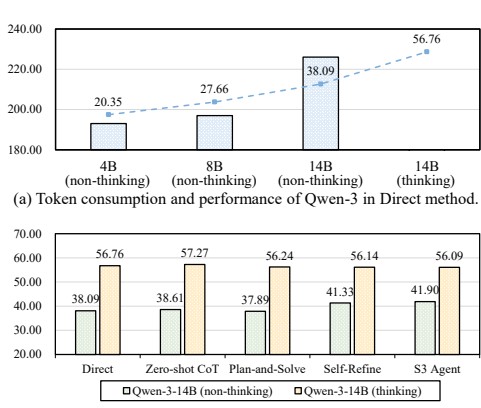

(a) Token consumption and performance of Qwen-3 in Direct method.

(b) Performance of different mode Qwen-3 in different methods.

Figure 3: Performance of different scale and thinking mode in Qwen-3. The line chart in (a) shows Direct performance of Qwen-3, while the histogram in (a) shows average output tokens across scales. The histogram in (b) shows performance in different methods.

LLMs adheres to the scaling law (Kaplan et al., 2020), larger models exhibit enhanced performance and longer output lengths. This suggests that more powerful LLMs engage in more extensive reasoning processes, resulting in superior thinking chain capabilities.

Additionally, we examine the performance of the Qwen-3-14B model with thinking mode activated and deactivated. Activating the thinking mode yields a significant improvement, with an average

| Source Language | English | | Chinese | | French | |
|---|---|---|---|---|---|---|
| Target Language | Chinese | French | English | French | English | Chinese |
| Meta-Llama-3.1-8B-Instruct | 19.88 ▲ | 14.00 ▼ | 23.35 ▲ | 13.38 ▼ | 17.30 ▲ | 18.58 ▲ |
| GPT-4o-mini | 39.47 ▲ | 38.53 ▼ | 38.19 ▲ | 35.03 ▲ | 32.05 ▲ | 28.84 - |
| Gemini-1.5-flash | 41.14 ▼ | 41.50 ▼ | 37.22 ▼ | 38.92 ▼ | 29.48 ▲ | 30.76 ▲ |
| Qwen-2.5-32B | 45.57 ▼ | 44.99 ▼ | 43.30 ▲ | 42.82 ▲ | 35.25 ▲ | 33.97 ▲ |
| DeepSeek-V3 | 62.55 ▲ | 60.88 ▲ | 61.31 ▲ | 59.36 ▲ | 57.69 ▲ | 53.84 ▲ |
| DeepSeek-R1-Distill-Qwen-32B | 50.79 ▼ | 51.45 ▼ | 50.85 ▲ | 48.90 ▲ | 52.56 ▲ | 51.28 ▲ |
| Qwen-3-14B | 57.91 ▲ | 57.03 ▼ | 53.77 ▼ | 58.39 ▲ | 51.28 ▲ | 50.00 ▲ |

Table 4: The results of CLP. ▲ presents the performance of CLP is better than the Direct method, ▼ presents the performance of CLP is worse than the Direct method.

increase of 16.9%. This indicates that RLLMs can achieve superior performance through extended reasoning processes.

### 4.8 IT'S IMPORTANT TO SELECT THE OPTIMAL ALIGNMENT LANGUAGE IN CLP

We explore the performance differences across various target languages within the CLP method, with the final results presented in Table 4. The findings indicate that different target languages yield varying performance gains, and directly aligning local languages to English through CLP is often not the most effective strategy. Utilizing various language alignments has demonstrated differing impacts on performance (Wang et al., 2025b), likely attributable to LLMs' comprehension of distinct language families.

Furthermore, different models utilize distinct optimal target languages identified by the CLP method, which may be attributed to varying language capabilities among the models. Although some researchers propose CLSP (Qin et al., 2023) and AutoCAP (Zhang et al., 2024a). CLSP investigates the integration of CLP and self-consistency methods to bridge the gap between different languages. AutoCAP automatically selects languages and automatic weight allocation to obtain the best language combination. This approach incurs substantial operational costs. Therefore, the selection of the best target language based on the multilingual capabilities of each model requires further exploration.

### 4.9 UNDERSTANDING THE ADVANTAGES OF LONG CoT

To elucidate the advantages conferred by Long CoT, we utilize GPT-4o-mini to systematically analyze the error types observed in Qwen-3-14B under both thinking and non-thinking modes, as illustrated in Figure 4.

The analysis reveals that, in the absence of the thinking mode, the reasoning error constitutes the predominant source of performance degradation. In contrast, activation of the thinking mode substantially mitigates such errors, highlighting the efficacy of Long CoT in enhancing the reasoning capabilities of RLLMs. Furthermore, RLLMs demonstrate superior comprehension of problem statements compared to LLMs, as evidenced by the fact that approximately one-third of LLM errors are attributable to misinterpretation of questions and underlying assumptions. These findings collectively underscore the critical role of Long CoT in advancing the overall performance of financial question answering models.

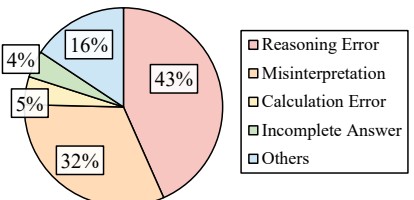

Figure 4: Error statistics for Qwen-3-14B under different thinking modes, considering cases where thinking mode is correct but non-thinking mode fails.

## 5 DISCUSSION

While considerable research has focused on enhancing the performance of LLMs across various domains, our experiments indicate that the key to improving LLM performance may lie in strength-

ening their long-term thinking capabilities, which aligns with the overarching objective of RLLMs. Consequently, we find that currently effective prompting methods and agentic frameworks for LLMs do not yield significant improvements for RLLMs. It is important to note that the agentic framework employed in our study was not specifically tailored for financial question answering. Thus, agentic frameworks that integrate RLLMs with techniques such as retrieval-augmented generation (RAG) and are well-designed for financial question answering may present a promising solution.

Furthermore, the Long CoT approach constrains the potential for enhancing performance of RLLMs through agentic frameworks. The advent of agentic reinforcement learning (Agentic RL) enables LLMs and RLLMs to function not only as standalone foundation models but also to be incorporated into complex reasoning and decision-making processes (Luo et al., 2025b; Zhang et al., 2025). Developing a suitable agentic workflow and further training LLMs and RLLMs through agentic reinforcement learning represents an intriguing avenue for improving model performance.

## 6 RELATED WORK

With the development of LLMs, more and more researchers focus on the application of LLM in various fields. Improving the performance of LLMs in downstream tasks has consistently been a focal point of researchers' efforts. Wei et al. (2022) explore the impact of Chain-of-Thought on LLMs and find that forcing thinking of LLMs can significantly improve performance. Inspired by this, researchers further explored the improvement of LLMs. Yao et al. (2023a) propose Tree-of-Thought (ToT), which enhances LLM by considering multiple reasoning paths and self-evaluating. Yin et al. (2023) introduce Exchange-of-Thought (EoT) to enable cross-model communication. In EoT, LLMs could enhance their performance through different network topologies. Zhang et al. (2024b) employs Multi-Perspective Verification and Wrong Information Utilization to prevent LLMs from repeating the same mistakes, thereby significantly enhancing their performance.

Some researchers have introduced complex agentic frameworks to enhance performance by standardizing the processes of LLMs. Yao et al. (2023b) present a novel agent framework ReAct. ReAct can track the interactions of LLMs with the environment and make decisions about the next steps. Li et al. (2023) propose a novel communicative agent framework that enables LLMs to understand tasks through role-playing. Hong et al. (2024) introduce Standardized Operating Procedures (SOPs) into frameworks and propose MetaGPT, which enhances the ability of software engineering in LLMs. Wang et al. (2024) propose $S^3$ Agent to improve the performance of LLMs on multi-modal problems by introducing multiple perspectives information.

## 7 CONCLUSION

In this work, we systematically investigate the factors influencing the performance of LLMs and RLLMs in financial question answering. Through extensive experiments on multiple models and methods, we find that the effectiveness of prompting strategies, agentic frameworks, and multilingual alignment approaches for LLMs largely stems from their ability to simulate longer reasoning chains (Long CoT). RLLMs, equipped with inherent Long CoT capabilities, show limited improvement from these conventional methods, highlighting a performance bottleneck. Our analysis suggests that future advances for RLLMs may require more sophisticated agentic mechanisms and dynamic reasoning processes. We hope these findings deepen our understanding of the Long CoT and serve as a valuable reference in financial question-answering.

### THE USE OF LARGE LANGUAGE MODELS (LLMS)

We declare that only LLMs were utilized to polish the English of this paper.

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

## A APPENDIX

### A.1 EXPERIMENTAL DETAILS

We use the default settings of the API platform. Here are the details of our experiments:

- Meta-Llama-3.1-8B-Instruct  temperature=1,top_p=1.
- GPT-4o-mini: temperature=1,top_p=1.
- Gemini-1.5-flash: temperature=1,top_p=1.
- Qwen-2.5-32B: temperature=.0.7,top_p=0.8.
- DeepSeek-V3: temperature=1,top_p=1.
- DeepSeek-R1-Distill-Qwen-32B: temperature=1,top_p=1.
- Qwen-3-14B (thinking): temperature=0.6,top_p=0.95.
- Qwen-3-14B8B4B (non-thinking): temperature=0.7,top_p=0.8.
- O4-mini: temperature=1,top_p=1.

