# OpenReview forum: "What Factors Affect LLMs and RLLMs in Financial Question Answering?"
_ICLR.cc/2026/Conference — ICLR 2026 Conference Withdrawn Submission_

### Official Review · Reviewer_RhS9 · 2025-10-15

**Soundness:** 2
**Presentation:** 2
**Contribution:** 2
**Rating:** 4
**Confidence:** 4

**Summary:**

This paper presents a detailed analysis of why LLMs and LRMs fail in financial question answering. The paper tests several models on various inference settings, e.g., direct prompt, CoT. They also experiment on the effectiveness of long-CoT and test on the variability between languages.

**Strengths:**

- This paper tests various models, including open-sourced and closed-sourced, which provide a comprehensive evaluation.
- This paper can provide a detailed analysis and shed light on the development of the domain-specific capability of LLMs

**Weaknesses:**

- All of the experiments are based on the FAMMA benchmark, which is multiple-choice, hindering the generality of the conclusion. More benchmarks are needed to validate the conclusion.
- The experimental results are common, e.g., Long-CoT yields better performance, and agent frameworks can help boost performance. And the authors don't propose any methods that can be used to reduce the errors made by LLMs in financial QA.
- typo, e.g., line 215 thing -> thinking

**Questions:**

See Above

---

### Official Review · Reviewer_6apo · 2025-10-23

**Soundness:** 2
**Presentation:** 3
**Contribution:** 2
**Rating:** 2
**Confidence:** 5

**Summary:**

This paper investigates how existing prompting strategies, agentic frameworks, and multilingual alignment methods affect the performance of standard LLMs and Reasoning LLMs on financial question answering. The authors evaluate five LLMs and three RLLMs on the FAMMA benchmark and report that (1) prompting and agent methods improve LLMs by simulating Long CoT; (2) conventional methods offer little benefit for RLLMs, which already exhibit strong intrinsic reasoning; and (3) multilingual alignment techniques mainly help LLMs by extending reasoning length but have minimal impact on RLLMs.

**Strengths:**

(1) The paper is clearly written and well-organized, with a logical experimental flow.
(2) The use of a standardized financial QA benchmark and a diverse set of models (including strong commercial and open-source systems) enhances reproducibility.
(3) The empirical observation that RLLMs are less responsive to conventional prompting may serve as a useful reference for practitioners.

**Weaknesses:**

(1) Lack of technical or conceptual novelty: The paper does not propose any new methods, frameworks, or theoretical insights. All evaluated techniques are directly borrowed from prior work.
(2) Limited depth in analysis: The paper reports performance trends but does not conduct ablations or controlled experiments to isolate why certain methods fail on RLLMs (e.g., is it due to overthinking, distribution shift in reasoning style, or redundancy?).  Without such analysis, the insights remain descriptive rather than explanatory.
(3) Empirical findings are largely expected: The core conclusions that LLMs benefit from structured prompting while RLLMs (which are trained or designed for extended reasoning) do not, are consistent with established intuitions about reasoning saturation and diminishing returns from external scaffolding. These observations, while valid, do not constitute a significant advance in understanding.

**Questions:**

(1) In the multilingual experiments, why were only three languages (EN, ZH, FR) included? Would the conclusions hold for truly low-resource languages?
(2) You suggest RLLMs “overthink” on easy questions. But is this due to excessive reasoning, or could it reflect distributional mismatch (e.g., RLLMs trained mostly on hard problems)? Can you show that manually limiting CoT length on easy questions improves RLLM accuracy?

---

### Official Review · Reviewer_U2V3 · 2025-10-26

**Soundness:** 2
**Presentation:** 3
**Contribution:** 2
**Rating:** 4
**Confidence:** 5

**Summary:**

This paper systematically analyzes how prompting, agentic frameworks, and multilingual alignment affect LLMs and RLLMs in financial QA.The key finding is that these methods boost standard LLMs primarily by simulating Long Chain-of-Thought (Long CoT) reasoning. In contrast, RLLMs, which inherently possess Long CoT, see diminished returns from these techniques, highlighting Long CoT as a current performance ceiling.

For LLMs, performance improves with longer, structured reasoning from prompts and agents.RLLMs are inherently robust; their gains come from agent collaboration, not longer CoT.Multilingual methods help LLMs via extended reasoning but offer minimal benefit to self-aligning RLLMs.

Future RLLM advancement may require complex, task-specific agents rather than longer reasoning chains.

**Strengths:**

1. The paper provides a comprehensive evaluation of various methods across both open-source and proprietary models, supported by substantial and robust experimental results.

2. It offers insightful conclusions on the key factors influencing performance and outlines a clear roadmap for future research.

**Weaknesses:**

1. Lack of discussion or experiments on the specificity of the financial domain. The paper uses only a single financial domain dataset for experiments, making it impossible to compare the effects of different models and methods on problems within versus outside the financial domain. The models used are all general-purpose models, lacking specialized financial domain models. This omission prevents the study from demonstrating how the same methods perform on domain-specific models. Furthermore, there is a lack of relevant concrete case analyses.
2. Issues with the novelty of conclusions: The paper dedicates significant space to arguing the impact of Long Chain-of-Thought (Long CoT) on model performance. In the conclusions, it claims: "Through extensive experiments on multiple models and methods, we find that the effectiveness of prompting strategies, agentic frameworks, and multilingual alignment approaches for LLMs largely stems from their ability to simulate longer reasoning chains (Long CoT)." Within this statement:
(1)The claim that "these methods simulate Long CoT" lacks sufficient citation or experimental substantiation.
(2)The assertion that "longer CoT improves performance" is arguably common knowledge in RLLM research, mentioned in technical reports from DeepSeek, Gemini, etc., and thus may not be sufficiently novel or a key finding for this paper.
3. Reproducibility issues: The experiments explore various prompting strategies and agent-related methods but fail to provide the specific prompt templates used. This omission hinders the reproducibility of the experiments.

**Questions:**

Questions
1. The study utilizes only a single financial dataset. Could the authors elaborate on whether the identified conclusions are specific to the financial domain, or if they are likely to be universally applicable across various reasoning domains?
2. Given the extensive body of research on Plan-and-Solve prompting and agentic frameworks, the paper evaluates just one representative method from each category. How can we be confident that the observed effects and conclusions hold true for the broader class of these methods, rather than being specific to the particular instances selected for this study?

Suggestion

For each method evaluated, please provide complete prompt examples in the appendix to enhance the reproducibility of the study.

---

### Official Review · Reviewer_NSJC · 2025-10-28

**Soundness:** 2
**Presentation:** 2
**Contribution:** 1
**Rating:** 2
**Confidence:** 4

**Summary:**

In the paper, the authors focused on evaluating the performance difference between large language models (LLMs) and reasoning large language models (RLLMs) under various setups, including prompt techniques, agent frameworks, and multilingual contexts. In the experiments, the authors present results evaluated on LLMs and RLLMs of various sizes, and derived evidence to support their claims and conclude that the ability to simulate longer reasoning chains is essential for performance improvement.

**Strengths:**

- The paper focuses on evaluating the empirical performance differences between LLMs and RLLMs across multiple dimensions, which offer good insights for the application in the financial domain.

- The clarity of the paper is good, with well-organized sections and a structure that clearly outlines the key observations and insights.

**Weaknesses:**

- The paper's breadth comes at the cost of depth and coherence. The topics covered appear compositional rather than complementary. It is not immediately clear how the performance of LLMs/RLLMs under various techniques (prompting, agents) meaningfully relates to their multilingual performance. The rationale for combining these two research directions in the same paper could be better articulated—specifically, how they echo each other or contribute to unified insights. The separate investigations might benefit from either deeper integration or independent exploration with greater depth in each area.

- In the experiments, some critical details are missing, and some of the claims are not well supported by the experiment results.
  - In the experiment section, the authors did not provide the details about how the agentic frameworks are implemented, or the details of the prompting techniques or examples.
  - In section 4.2, the authors claim that an effective prompting method significantly enhances the performance of LLMs. However, Table 1 shows that prompting techniques lead to worse performance in some subtask categories, and even when examining the aggregated overall performance, the improvements are marginal rather than significant.
  - In section 4.3, the authors claim that the smaller LLMs benefit more from the agentic framework. But in Table 1, only one small model is evaluated, and I think more evidence is needed to show whether this behavior is unique to the model or a more general behavior.
  - In section 4.4, the authors claim that the reading length significantly enhances the performance of LLMs, with the evidence of the performance of the model with CLP. First, as the method and the reasoning length are changed simultaneously, it would be hard to attribute which part contributes more to the performance difference. And the conclusion would be clearer if the measurement of reasoning length were included in the analysis.
  - In section 4.3, the authors claim the performance gains of RLLMs in the agentic framework are due to agent collaboration. However, there is no metric/measurement introduced to measure agent collaboration or a concrete analysis to support their claim.

- Many of the findings have been well-established in prior work, which limits the paper's significance. For example, the performance improves as LLMs' parameters increase, which is well understood in the context of the scaling law; the longer reasoning steps generally lead to better performance, which is also well understood and studied comprehensively in previous works, e.g., [1]

[1]: Jin, M., Yu, Q., Shu, D., Zhao, H., Hua, W., Meng, Y., ... & Du, M. (2024). The impact of reasoning step length on large language models. arXiv preprint arXiv:2401.04925.

**Questions:**

Please see the above section.

---

### Note · Authors · 2025-11-12

**Comment:**

We thank all reviewers for their insightful and thoughtful feedback. Based on the reviewers' comments, we have decided to withdraw this paper and further refine it. Thank you again to all reviewers and PC's efforts.

**Withdrawal Confirmation:**

I have read and agree with the venue's withdrawal policy on behalf of myself and my co-authors.